# A Case of Light Chain Deposition Disease Leading to Acute Liver Failure and Review of Literature

**DOI:** 10.3390/diseases11010024

**Published:** 2023-02-01

**Authors:** Mustafa Gandhi, Syed Bilal Pasha, Emily Reznicek, Syed Raheel Pasha, Hamza Ertugrul, Adel Araslanova, Feng Yin, Veysel Tahan

**Affiliations:** 1Department of Internal Medicine, University of Missouri, Columbia, MO 65211, USA; 2Division of Gastroenterology and Hepatology, Department of Internal Medicine, University of Missouri, Columbia, MO 65211, USA; 3Department of Medicine, Jinnah Sindh Medical University, Karachi 75510, Pakistan; 4Department of Medicine, University of South Florida, Tampa, FL 33620, USA; 5Department of Pathology, University of Missouri, Columbia, MO 65211, USA; 6Department of Pathology, The University of Texas MD Anderson Cancer Center, Houston, TX 77030, USA

**Keywords:** light change deposition disease, acute liver failure, acute kidney injury, plasma cells, monoclonal, immune, gammopathy

## Abstract

Light chain deposition disease (LCDD) is a monoclonal immunoglobulin deposition disease characterized by light chain deposition in soft tissues and viscera, causing systemic organ dysfunction with an underlying lymphoproliferative disorder. While the kidney is the most affected organ, cardiac and hepatic involvement is also seen with LCDD. Hepatic manifestation can range from mild hepatic injury to fulminant liver failure. Herein, we are presenting a case of an 83-year-old woman with a monoclonal gammopathy of undetermined significance (MGUS), who presented to our institution with acute liver failure progressing to circulatory shock and multiorgan failure. After an extensive workup, a diagnosis of hepatic LCDD was determined. In conjunction with the hematology and oncology department, chemotherapy options were discussed, but given her poor prognosis, the family decided to pursue a palliative route. Though establishing a prompt diagnosis is important for any acute condition, the rarity of this condition, along with paucity of data, makes timely diagnosis and treatment challenging. The available literature shows variable rates of success with chemotherapy for systemic LCDD. Despite chemotherapeutic advances, liver failure in LCDD indicates a dismal prognosis, where further clinical trials are difficult owing to the low prevalence of the condition. In our article, we will also be reviewing previous case reports on this disease.

## 1. Introduction

First described in 1976 by Randall et al. [1], monoclonal immunoglobulin deposition disease (MIDD) is a rare condition characterized by the widespread deposition of immunoglobulins, excessively produced by monoclonal plasma and B cells [2,3]. An underlying lymphoplasmacytic neoplasm or plasma cell dyscrasia, such as multiple myeloma or a monoclonal gammopathy of undetermined significance (MGUS), is often seen [3]. Though light chain deposition is the most common variant; mixed and heavy chains can also be seen [3]. Therefore, MIDD is categorized into four classes:Heavy chain deposition disease (HCDD).Light chain deposition disease (LCDD).Heavy and light chain deposition disease (HLCDD).Immunoglobulin light chain (AL) amyloidosis.

The majority (80–90%) of immunoglobulin light chains (LCs) in LCDD cases are kappa in configuration and accumulate in the soft tissues and viscera as granular deposits, leading to progressive organ dysfunction. Male predominance (60–65%) is seen, and the median age of presentation is 50–60 years [2,4]. Although the exact incidence is uncertain, extrapolations of studies suggest that a few hundred patients are annually diagnosed in the United States [5].

Given that MIDD is a systemic disorder, the clinical presentation can vary depending on the organ involvement and severity of dysfunction. Kidneys are, by far, the most involved organ, followed by the heart and the liver. Spleen, peripheral nervous system (PNS), skin and other organ involvement are less frequent [2]. Isolated extrarenal involvement is rarer.

Here, we present a case of acute liver failure due to LCDD in an elderly woman with an underlying monoclonal gammopathy of undetermined significance.

## 2. Case Presentation

An 83-year-old woman with a history of a monoclonal gammopathy of undetermined significance (MGUS) was admitted to our hospital for shortness of breath and anasarca. Her only medication was a multivitamin, and there was no history of substance or alcohol use. Her vital signs were within the normal range. The physical exam was notable for lower extremity pitting edema extending to the thighs. No hepatosplenomegaly was noted. The labs were significant for a total bilirubin of 2.49 mg/dL, alkaline phosphatase of 590 U/L, AST of 149 U/L, ALT of 55 U/L, INR of 1.2, and sodium of 133 mg/dL. Her kidney function was normal, with a creatinine level of 0.8 mg/dL. The serologies for hepatitis A, B, C and HIV were negative, but Epstein Barr virus (EBV) IgG and IgM viral capsular antigen (VCA), and EBV nuclear antigen were positive, and cytomegalovirus DNA was detected in the serum at 104 IU/L. A test for leptospirosis was not performed, owing to a low clinical suspicion.

An abdominal ultrasound showed a right pleural effusion and normal appearances of the liver and bile ducts. An echocardiogram showed moderate to severe aortic stenosis without regurgitation, a left ventricle ejection fraction of 72% with a grade II diastolic dysfunction, and an estimated RVSP of 47 mmHg. Left cardiac catheterization was performed, which revealed an elevated LVEDP of 20–25 mmHg, suggestive of acute decompensated heart failure. The acute liver dysfunction was believed to be multifactorial in the setting of a new heart failure diagnosis with a superimposed EBV infection. Diuresis was initiated in the hospital, and she was discharged to return home on oral diuretics.

The patient returned to the hospital 1 week later with worsening jaundice and encephalopathy. Her vital signs were within the normal range, and the physical exam was significant for anasarca and jaundice. Her labs demonstrated the worsening of her liver function, with a total bilirubin of 16.25 mg/dL, direct bilirubin of 13.5 mg/dL, alkaline phosphatase of 1116 U/L, AST of 289 U/L, ALT of 95 U/L, INR of 1.3, albumin 3.2 of g/dL, serum sodium of 127 mmol/L and an acute kidney injury with creatinine at 1.3 mg/dL. The urinalysis was negative for proteinuria. The procalcitonin level was elevated at 7.68 ng/mL, which was thought to be falsely positive owing to shock or an unknown underlying infection. An MRI of the abdomen showed the normal contour of the liver without a mass, thrombosis, or biliary obstruction. The features concerning for acute cholangitis, such as biliary ductal dilation or an increased signal intensity around the bile ducts on T2-weighted images, were not seen. The additional labs, including an iron panel, ceruloplasmin and antibodies associated with autoimmune hepatitis (antismooth muscle antibody, liver kidney microsome type 1, antimitochondrial antibody), were all unremarkable.

Despite treating her with diuresis, broad spectrum antibiotics, lactulose and rifaximin, her liver and kidney function continued to worsen, and owing to diagnostic uncertainty, a liver biopsy was deemed necessary. An ultrasound-guided transcutaneous biopsy of the left lobe of liver was obtained on day 4 of hospitalization. Unfortunately, the patient progressed to acute liver failure and circulatory shock while awaiting her biopsy results and was transferred to the intensive care unit (ICU). In the ICU, the workup for circulatory shock was unremarkable. The infectious workup, including chest radiograph, blood, urine cultures and respiratory viral pathogen panels, were negative for infection. Her serum cortisol was normal as well. The repeat echocardiogram was unchanged from the prior one and demonstrated the adequate inspiratory collapse of the inferior vena cava. The etiology was determined to be multifactorial, and intravenous vasopressors, including vasopressin, phenylephrine and norepinephrine, were administered to maintain adequate perfusion. By day 10 of her admission, bilirubin had reached 36.5 mg/dL, coagulopathy worsened (INR 3.6) and serum creatinine was unmeasurable owing to icteric samples. Previous trends showed the progressive worsening of serum creatinine (maximum of 2.79 mg/dL) with anuria, significant proteinuria and hematuria. The etiology of this anuric renal failure was thought to be a combination of ischemic acute tubular necrosis and the renal manifestation of LCDD.

Given the patient’s decline, the pathology slides were expedited for interpretation. A review of the slides showed diffuse dense eosinophilic hyaline material expanding the sinusoid and compressing the hepatocyte plate, morphologically resembling amyloidosis (Figure 1). There was no significant portal or lobular inflammatory infiltration. A scattered bile pigment was noted within hepatocytes, consistent with hepatocellular cholestasis. There was no evidence of advanced fibrosis on the trichrome stain. Upon staining, the sinusoidal eosinophilic hyaline materials were PAS positive and Congo red negative and did not show birefringence under polarized light. These findings were consistent with the diagnosis of monoclonal immunoglobulin deposition disease. The serologies showed glycosylated M-protein light chain on serum protein electrophoresis (SPEP) and an elevated kappa/lambda free light chain ratio of 16.6. The hematology and oncology department was consulted, and the experts there agreed that the clinical picture was consistent with LCDD. Chemotherapy options were discussed, but given her multiorgan system failure, the patient’s power of attorney decided to discharge her with hospice care.

## 3. Discussion

As mentioned above, MIDD, including LCDD, is a consequence of the systemic deposition of fragmented immunoglobulin chains produced by abnormal plasma cells and lymphocytes. Therefore, disorders characterized by abnormal clonal proliferation of these cells, such as multiple myeloma (11–65%), MGUS (32–86.8%) and macroglobulinemia (2%), are often concurrently present [3]. A muticenter retrospective study by Pozzi et al. found that in up to 32% of cases, no such disorders can be identified. These patients are classified as having idiopathic LCDD [2]. The characteristics of the previously described cases of LCDD with hepatic involvement have been summarized in Table 1.

The deposition of kappa or lambda light chains (LCs) is a hallmark of LCDD [3]. The immunoglobulin fragments are nonamyloid and do not possess a fibrillar ultrastructure, which is responsible for its inability to stain with Congo red [6] Notably, a Congo red stain is the key to differentiate LCDD from AL amyloidosis, a condition in which the deposition is also composed of monoclonal light chains and highlighted by a positive Congo red stain. They can deposit in multiple organs, such as the kidneys, liver, peripheral nerves or the heart [2] Because it is a systemic disorder, the clinical presentation of LCDD is determined mostly by the organ involved. Kidneys are by far the most affected organ (90–96%), resulting in renal dysfunction [2,7] In our patient, although there was a mild renal injury at admission, the progressive renal function decline during the course of the hospital stay was believed to be multifactorial, instead of exclusively from LCDD. Cardiac involvement is also commonly seen in LCDD, which can partially explain the finding of the left ventricular hypertrophy with diastolic dysfunction in our patient, but a lack of tissue diagnosis made it impossible to confirm.

After the heart, the liver is the most affected extrarenal organ [2], associated with varying degrees of hepatic dysfunction [8,9,10,11,12,13,14,15,16,17] There is usually concomitant renal involvement in those with severe liver disease. In those with established liver failure, the prognosis is guarded [10,11]. As seen with other causes of liver failure, the clinical presentation in these patients is generally encephalopathy, coagulopathy, and jaundice. Very similar to our case, a predominantly cholestatic pattern of liver injury with hepatic LCDD has previously been observed by Mena-Durán [11] and Faa et al. [15].

The diagnosis of organ dysfunction can be made by routine laboratory work, depending on the affected system. Renal involvement usually manifests as renal failure and nephrotic range proteinuria. When the deposits affect the kidneys, metabolic alkalosis, and hypokalemia are seen, owing to the loss of potassium and hydrogen when the renal tubules are affected [18] Hepatic dysfunction typically manifests as an elevated AST and ALT with disproportionately high alkaline phosphatase, jaundice, and hepatomegaly. The predominantly cholestatic pattern of a liver injury has been proposed to occur owing to intrahepatic biliary stasis, as a consequence of extrinsic compression from portal deposits, but the exact mechanism remains unclear [15] Our patient’s liver biopsy showed light chain deposits expanding the sinusoids, with associated canalicular and hepatocellular cholestasis. Most cases in the literature have demonstrated similar histopathology, with some differences. Plummer et al. [9] described scattered sinusoidal eosinophilic deposits, while Tsushima et al. [19] discovered hepatocyte atrophy and perisinusoidal deposits. We did not perform immunohistochemical staining in our case, but where it has been performed, the staining was positive for anti-kappa (most commonly) or antilambda light chains in the sinusoidal deposits [12,20,21].

As the disease progresses, a decline in hepatic synthetic function can be shown by hypoalbuminemia and coagulopathy, along with the development of portal hypertensive ascites. In rare cases, it can progress to acute liver failure, which presents as encephalopathy and jaundice. In prior cases of hepatic LCDD, various degrees of liver injury are seen, but our case appears to be the most severe, as shown by the severe hepatic enzyme derangement, ultimately progressing to liver failure.

The acute liver failure (ALF) seen in our patient can be seen with other etiologies, such as acute viral hepatitis, Budd–Chiari syndrome, acetaminophen overdose, autoimmune hepatitis, drugs and ischemic liver failure [22]. In the case of our patient, other factors that could concomitantly cause liver disease were excluded by negative serological tests for viral hepatitis. Metabolic disorders such as hemochromatosis, Wilson’s disease and autoimmune hepatitis were ruled out by appropriate testing. Although the patient had a partial portal vein thrombosis, an MRI scan ruled out Budd–Chiari syndrome. Additionally, acetaminophen overdose and other hepatotoxic medications were ruled out on history. Although viral serologies could be negative in the serological window period and although multivitamins, which our patient was taking, can lead to drug-induced liver injury (DILI), the liver biopsy ultimately helped rule out these etiologies and established the diagnosis of LCDD. Furthermore, despite the discontinuation of multivitamins on hospitalization, the patient continued having worsening liver failure, providing further evidence. Lastly, DILI from vancomycin and piperacillin-tazobactam was unlikely because the liver injury was present before the initiation of these antibiotics. Adequate perfusion via vasopressors was also maintained, making shock liver less likely. When compared to previous cases of LCDD with hepatic involvement, the clinical decline in our patient was steeper with ensuing multiorgan failure. A reduction in hepatic blood flow from a portal thrombosis and aortic stenosis, as well as CMV and EBV infectious states, might have contributed to the liver dysfunction and overall severity of disease. Another unique aspect of our case was the presence of MGUS, whereas all other cases have reported multiple myeloma as an underlying lymphoproliferative disorder [10,11,23] It is unclear how the MGUS could have contributed to the rapid progression of disease seen in our patient.

**Table 1 diseases-11-00024-t001:** Characteristics of reported cases of LCDD with hepatic involvement.

	Year	Age	Gender	Race	Symptoms	Total Bilirubin	Albumin	AST (U/L)	ALT (U/L)	Alkaline Phosphatase (U/L)	INR	Creatinine	Hepatomegaly	Ascites	Encephalopathy	Treatment	Outcome	Lymphoproliferative Disorder	Liver Biopsy
**Our study**	2022	83	F	Caucasian	Generalized edema	36.45 mg/dL	3.2 g/dL	289	95	1116	3.6	2.79 mg/dL	No	Yes	Yes	Supportive, CTX offered but declined	Multiorgan failure, circulatory shock; died shortly after	Monoclonal gammopathy of unknown significance	PAS positive, Congo red negative pale pink hyaline material expanding the sinusoid throughout the biopsy
**Plummer et al.** [9]	2021	42	M	Unknown	Abdominal pain, fatigue, early satiety	0.9 mg/dL	2.2 g/dL	91	118	990	Unknown	Unknown	Yes	Yes	Unknown	Bortezomib, cyclophosphamide, and dexamethasone	Discontinued CTX and died shortly thereafter	Unknown	Scattered sinusoidal eosinophilic deposits; lambda restriction on IF
**Talukdar et al.** [8]	2013	55	M	Unknown	Dyspnea, edema, JVD, muscle wasting	Unknown	2 g/dL	Unknown	Unknown	822	Unknown	2 mg/dL	Yes	Yes	Unknown	Bortezomib-based induction CTX offered but patient declined	Lost to follow-up	Unknown	Extracellular perisinusoidal deposits compressing bile ductules, hepatocytes and portal tracts; it also showed Kupffer cell hyperplasia
**Tsushima et al.** [19]	2021	64	F	Japanese	Malaise and anorexia	Unknown	3.8 g/dL	13	21	Unknown	Unknown	6.31 mg/dL	Unknown	Yes	Unknown	Daratumumab, bortezomib and dexamethasone	Responded to treatment	Ig D Multiple myeloma	Hepatocyte atrophy and perisinusoidal space deposits; IF positive for antikappa
**Mena-Durán et al.** [11]	2011	81	M	Caucasian	Jaundice, anorexia, weight loss, macroglossia	6.35 mg/dL	1.9 g/dL	102	33	699	Unknown	1.3 mg/dL	Yes	Unknown	Yes	High-dose dexamethasone	Died within 3 weeks of hospitalization	IgG Kappa Multiple Myeloma	Perisinusoidal deposits of Congo red negative and eosinophilic material. Kappa chain restriction on IF
**Cristino et al.** [16]	2017	60	M	Caucasian	Weakness, dyspnea on exertion, early satiety	Elevated	Unknown	Unknown	Unknown	1377	Unknown	2.7 mg/dL	Yes	Unknown	Unknown	Chemotherapy (not specified)	Died 2 months later with infectious complication	No	Sinusoidal deposits of eosinophilic and Congo red negative material. IHC with antikappa Ab inconclusive
**Grembiale et al.** [20]	2020	70	M	Unknown	Fatigue and weight loss	4.8 mg/dL	Unknown	647	485	Unknown	Unknown	6.7 mg/dL	Yes	Unknown	Unknown	Dexamethasone andbortezomib	Complete recovery in 6 months	Unknown	Congo red and PAS negative, eosinophilic deposits; kappa LC predominant on IF
**Kwon JH et al.** [12]	2011	62	M	Mongolian	Generalized edema, hepatomegaly	0.4 mg/dL	2 g/dL	76	53	276	Unknown	1.7 mg/dL	Unknown	Yes	Unknown	Peg-IFN + ribavirin for coexisting chronic HCV	Died 2 months later	Multiple myeloma	Sinusoidal expansion with eosinophilic, Congo red negative deposits; IHC stains show deposits composed of kappa LC
**Michopulous S et al.** [10]	2002	36	M	Unknown	Jaundice	7.8 mg/dL	Unknown	89	100	2315	Unknown	Unknown	Yes	Unknown	Unknown	Unknown	Rapid decline with multiorgan failure and death	Multiple myeloma	Slightly eosinophilic, Congo red negative, PAS positive perisinusoidal deposits; positive for kappa LC on IF
**Kumar PN et al.** [13]	2012	66	M	Unknown	Jaundice and pruritis	11.3 mg/dL	2.7 g/dL	177	75	1965	Unknown	Unknown	Yes	Unknown	Unknown	Unknown	Progressive worsening and loss to follow-up	Unknown	Congo red negative, PAS positive, eosinophilic perisinusoidal deposits compressing hepatocytes; bile ductules
**Pelletier et al.** [21]	1988	63	F	Unknown	Weight loss	44 µmol/L	47 IU/L	Unknown	Unknown	469	Unknown	71 µmol/L	Yes	Unknown	Yes	Cyclophosphamide, melphalan andprednisolone × 6 months	Died in 9 months	Unknown	Deposition of Congo red negative, antilamba LC positive material along the space of Disse outlining the sinusoids
**Girelli et al.** [14]	1998	59	F	Caucasian	Fatigue, anorexia, epigastric pain, weight loss	Unknown	Unknown	Unknown	Unknown	632	Unknown	Unknown	Yes	Unknown	Unknown	Received at different center	Lost to follow-up	Unknown	Slightly eosinophilic, PAS positive, Congo red negative, staining positively with antikappa antiserum in the perisinusoidal area
**Faa G et al.** [15]	1989	67	M	Unknown	Jaundice	11.4 mg/dL	2.12 g/dL	94	87	3905	Unknown	Unknown	Yes	Unknown	Unknown	CTX (not specified) following 32 mg methylprednisolone	Improved	Unknown	Congo red negative, PAS positive, eosinophilic deposits in the space of Disse, making sinusoidal appear thickened; Congo red positive deposits were seen in arterial wall and connective tissue; incubation with antilamba LC antiserum showed intense staining and specific staining with antikappa
**Ichikawa et al.** [23]	2007	62	F	Unknown	Facial edema and lumbar pain	3.2 mg/dL	Unknown	Unknown	Unknown	Unknown	Unknown	3.9 mg/dL	Unknown	Unknown	Unknown	Dexamethasone, plasma exchange and dialysis	Death from disseminated intravascular coagulation	Multiple myeloma	Amyloid-like deposits in perisinusoidal space, Congo red negative, IgA and kappa positive on immunohistochemical stain
**Bedossa P et al.** [17]	1988	54	M	Unknown	Hepatomegaly and portal hypertension	Unknown	Unknown	Unknown	Unknown	Very elevated	Unknown	Unknown	Yes	Unknown	Unknown	Unknown	Hepatic failure, renal failure	Unknown	PAS positive, Congo red negative, amorphous ribbon-like material, green with Masson’s trichrome and red after picrosirius
**Bedossa P et al.** [17]	1988	69	M	Unknown	Hepatomegaly and splenomegaly	Unknown	Unknown	Unknown	Unknown	Elevated	Unknown	Unknown	Yes	Unknown	Unknown	Unknown	Renal failure, portal hypertension	Myeloma	PAS positive, Congo red negative, amorphous ribbon-like material, green with Masson’s trichrome and red after picrosirius
**Bedossa P et al.** [17]	1988	60	F	Unknown	Hepatomegaly	Elevated	Unknown	Unknown	Unknown	Very elevated	Unknown	Unknown	Yes	Unknown	Unknown	Unknown	Portal hypertension, renal abnormalities	Unknown	PAS positive, Congo red apple green birefringence, amorphous ribbon-like material expanding the sinusoid, green with Masson’s trichrome and red after picrosirius

IF: immunofluorescence; PAS: periodic acid–Schiff; LC: Light chain; JVD: jugular vein distention.

As with our case, LCs in hepatic LCDD are deposited in a perisinusoidal pattern that contrasts with amyloidosis, wherein a mostly parenchymal pattern of deposition is seen [13]. Immunofluorescence (IF) microscopy can often detect immunoglobulin light chains and C1q and C3 complement proteins [17] Electron microscopy demonstrates punctate granular deposits that can be diagnostic. Most of the immunofluorescence and electron microscopy findings have been described in the kidneys because they are the most involved organs, but the characteristics of the light chain deposits remain similar in other organs [17].

Most cases of LCDD also have an underlying hematological malignancy, such as multiple myeloma, MGUS or another plasma cell dyscrasia, leading to the increased light chain secretion. The disorder may have been previously diagnosed or discovered at the time of presentation for the LCDD. The lymphoproliferative disorder can be worked up with flow cytometry, serum protein electrophoresis and light chain fractions with or without a bone marrow biopsy. Data from other plasma cell dyscrasias argue for the use of proteosome inhibitor bortezomib and high-dose melphalan for the treatment of LCDD [24,25] Previous case reports of systemic LCDD with hepatic involvement have also demonstrated clinical improvement and recovery with a Bortezomib-based chemotherapy regimen, in addition to systemic glucocorticoids [19,20] Another agent that has shown promise in refractory cases is the monoclonal antibody directed against CD-38, daratumumab [19,26] However, even with the recent advancement in chemotherapy, once hepatic failure sets in, the prognosis remains extremely poor [27], demanding a high degree of suspicion and necessitating early diagnosis. For our patient, treatment with a cyclophosphamide, bortezomib and dexamethasone (Cy-BorD) regimen was proposed by the hematology and oncology department, but because of our patient’s multiorgan dysfunction, her overall poor prognosis and her likely inability to tolerate or have a meaningful clinical response with chemotherapy, it was decided per her family’s wishes to instead pursue palliative care.

At present, supportive therapy remains the mainstay of management for such patients. Dialysis can improve survival with renal failure. Liver failure from LCDD is managed like liver failure from any other cause. This includes albumin replacement, lactulose therapy, antibiotics and diuresis, as was the case with our patient. In end-stage organ disease, transplantation may become necessary. This is a rare condition with a very poor prognosis and limited treatments. It is essential for clinicians to be vigilant for renal and liver dysfunction, especially in patients with underlying lymphoproliferative disorders such as multiple myeloma and MGUS.

## 4. Conclusions

LCDD is a rare condition that classically presents with renal disease, but clinicians should be aware of the extrarenal manifestations of LCDD. The heart and liver are two of the most common extrarenal organs involved. Hepatic manifestations of LCCD can at times present with acute liver failure. which is generally associated with an extremely poor prognosis. The treatment options for such patients, outside of supportive measures, are limited and directed mostly toward underlying plasma cell disorders. Among broad differentials for acute liver failure, hepatic LCDD should also be kept in mind, and a prompt liver biopsy should be obtained in patients with diagnostic uncertainties. Bortezomib-based chemotherapy, along with systemic steroids and anti-CD38 antibody daratumumab, has shown encouraging results and can be considered in such patients.

## Figures and Tables

**Figure 1 diseases-11-00024-f001:**
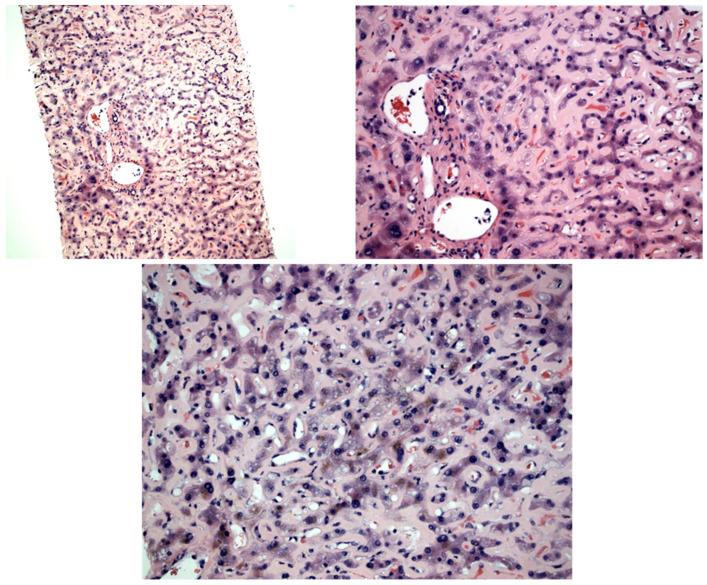
H&E (3A, 100×; 3B and 3C, 200×) shows dense eosinophilic hyaline material expanding the sinusoid throughout the biopsy. Focal yellow bile pigment can be seen, consistent with hepatocellular cholestasis. There is no significant portal or lobular inflammation. The eosinophilic hyaline material is negative on Congo red stain.

## Data Availability

Data sharing not applicable. No new data were created or analyzed in this study. Data sharing is not applicable to this article.

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
