# Peer review of "A Case of Light Chain Deposition Disease Leading to Acute Liver Failure and Review of Literature"

_diseases, 2023, doi:10.3390/diseases11010024_

Round 1

Reviewer 1 Report

The report of A case of Light chain deposition disease leading to acute liver failure and review of literature is very interesting because it allows us to see the approach taken in a patient with these characteristics

Author Response

Minor spelling/grammatical issues fixed

Reviewer 2 Report

In this manuscript, the authors aimed to report a case of hepatic LCDD with multi-organ failure, which is quite different from previous cases and worth discussing. However, the following concerns should be addressed.

Major comments

1. Cases of hepatic LCDD without renal involvement have been previously reported. Could the authors highlight in the discussion section the differences between this case and previous ones, e.g., the combination of MGUS, the multi-organ failure, and the more rapid progression, and discuss possible causes for these differences?

2. Is it possible to present and analyze the liver pathology findings in more detail and compare the differences with the pathological presentation in earlier cases?

3. Sadly, the patient received no further treatment in this case. However, the implications of this case for more rapid diagnosis and follow-up of patients with previous related hematologic disorders can be discussed.

Also, there exist some minor issues, on page 1 line 16, “is also seen” is repeated.

Author Response

Cases of hepatic LCDD without renal involvement have been previously reported. Could the authors highlight in the discussion section the differences between this case and previous ones, e.g., the combination of MGUS, the multi-organ failure, and the more rapid progression, and discuss possible causes for these differences?

When compared to previous cases of LCDD with hepatic involvement, the clinical decline in our patient was steeper with ensuing multiorgan failure. A unique aspect about our case was the MGUS while all other cases report multiple myeloma if an underlying lymphoproliferative disorder was present. It is unclear how the MGUS could have contributed to the rapid progression of disease seen in our patient. One factor contributing to the severity of the illness could be additional ischemic injury to the liver from portal thrombosis reducing hepatic blood flow as well as the recently diagnosed aortic stenosis. The CMV and EBV infectious states might also add to the liver dysfunction.

Is it possible to present and analyze the liver pathology findings in more detail and compare the differences with the pathological presentation in earlier cases?

The histopathological study of the liver biopsy showed pale, pink hyaline material expanding the sinusoids. These materials were PAS stain positive, Congo-Red stain negative with associated canalicular and hepatocellular cholestasis and normal trabecular architecture on reticulin stain. Most cases on literature review present with similar histopathology with some deviations. Plummer et al. describe scattered sinusoidal eosinophilic deposits, while Tsushima et al. discovered hepatocyte atrophy and perisinusoidal deposits on histopathological studies. We did not perform immunohistochemical staining in the tissue sample but in cases where it was performed the staining was positive for anti-kappa (most commonly) or anti-lambda light chains in the sinusoidal deposits.

Sadly, the patient received no further treatment in this case. However, the implications of this case for more rapid diagnosis and follow-up of patients with previous related hematologic disorders can be discussed.

This is a rare condition with a very poor prognosis and limited treatments. It is essential for clinicians to be vigilant for renal and liver dysfunction especially in patients with underlying lymphoproliferative disorders such as multiple myeloma and MGUS.

Also, there exist some minor issues, on page 1 line 16, “is also seen” is repeated.

Fixed

Reviewer 3 Report

In the manuscript ‘A case of Light chain deposition disease leading to acute liver  failure and review of literature’, Authors describe the clinical case of acute liver failure secondary to Light chain deposition disease.

In my opinion, the case is extremely interesting. The manuscript presentation is good. However, to my knowledge, cases of liver involvement and progressive, but not acute, organ failure are reported in the literature. Therefore, concomitant factors and/or co-factors capable of uncovering organ damage secondary to LCDD must be excluded for a correct diagnosis. Therefore some clarifications are needed.

In particular:

·         It is necessary to add in the discussion the differential diagnosis between pathologies with overlapping clinical picture. For example, mention Budd-Chiari syndrome for the possible acute presentation and characteristics similar to the case in question (the Authors report no thrombosis on abdominal MRI)

·         It is necessary to explain how (and if) damage factors or co-factors were excluded. In particular: how were acute viral hepatitis excluded? Only serology? Have viraemias been tested? The antibody status could in fact be falsely negative at an early stage.

·         In this regard, the Authors report positivity for EBV IgM and IgG. What is it referring to? VCA or EBNA? Was viraemia tested?

·         Was the patient evaluated for Leptospirosis? What was the value of the platelets?

·         Has cholangitis been ruled out? What was the value of procalcitonin?

·         How was possible iatrogenic etiology (DILI) ruled out? The patient was being treated with multivitamins.

Moreover appears useful to:

·       Specify whether the liver biopsy was performed transcutaneosly or transjugularly

·       Modify the table (16?), now unreadable.

For these reasons, I would consider this study suitable for publication in “Diseases” after major revisions.

Author Response

It is necessary to add in the discussion the differential diagnosis between pathologies with overlapping clinical picture. For example, mention Budd-Chiari syndrome for the possible acute presentation and characteristics similar to the case in question (the Authors report no thrombosis on abdominal MRI)

The hepatic failure which was seen in our patient can be seen with other etiologies causing acute liver failure (ALF) such as acute viral hepatitis, Budd-Chiari syndrome, acetaminophen overdose, autoimmune hepatitis, drugs, and ischemic liver failure. Although the patient had partial portal vein thrombosis, an MRI scan ruled out Budd-Chiari syndrome. Similarly appropriate testing for viral hepatitis, excluding acetaminophen overdose and liver injury casing drugs, and testing for antibodies associated with autoimmune hepatitis excluded some of the other causes of the liver failure. The patient was treated with vasopressors to maintain adequate perfusion in the setting of shock. It is necessary to rule out confounding causes of ALF in such cases for appropriate medical management.

It is necessary to explain how (and if) damage factors or co-factors were excluded. In particular: how were acute viral hepatitis excluded? Only serology? Have viraemias been tested? The antibody status could in fact be falsely negative at an early stage.

In the case of our patient, other factors which could concomitantly cause liver disease were excluded by testing for viral hepatitis based on serological testing, metabolic disorders such as hemochromatosis, Wilson’s disease, and autoimmune hepatitis. Although viral serologies could be negative in the serological window period, the liver biopsy ultimately clinched the diagnosis of LCDD

In this regard, the Authors report positivity for EBV IgM and IgG. What is it referring to? VCA or EBNA? Was viraemia tested?

The patient tested positive for Epstein Barr Virus (EBV) IgG and IgM viral capsular antigen (VCA) and EBV nuclear antigen. Cytomegalovirus DNA was detected in the serum at 104 IU/L.

Was the patient evaluated for Leptospirosis? What was the value of the platelets?

The patient was not tested for leptospirosis due to a low clinical suspicion and normal platelet count of about 300 x 109/L. 

Has cholangitis been ruled out? What was the value of procalcitonin?

MRI of the abdomen did not show any features concerning for acute cholangitis such as bile duct dilation or an increase in signal intensity around the bile duct on T2-weighted images and the procalcitonin level was elevated at 7.68 ng/ml which was thought to be falsely positive due to shock or an unknown underlying infection

How was possible iatrogenic etiology (DILI) ruled out? The patient was being treated with multivitamins.

The patient was on multivitamin supplements which can lead to drug-induced liver injury (DILI). Despite the discontinuation of multivitamins on hospitalization, the patient continued having worsening liver failure with the biopsy histopathology ultimately clinching the diagnosis. DILI from vancomycin and piperacillin-tazobactam was unlikely because the liver injury was present before initiation of these antibiotics.

Specify whether the liver biopsy was performed transcutaneosly or transjugularly

Ultrasound guided transcutaneous biopsy of the left lobe of liver was performed.  

Modify the table (16?), now unreadable.

Table modified

Round 2

Reviewer 3 Report

In my opinion concomitant factors (probably infectious) were the trigger of the acute liver failure in this case report. In any case, the clinical case remains interesting above all in the light of the histological data. Authors have responded satisfactorily to the requested revisions.